# Unbiased Contrastive Divergence Algorithm for Training Energy-Based Latent Variable Models

**Yixuan Qiu**
Department of Statistics and Data Science
Carnegie Mellon University
Pittsburgh, PA 15213, USA
yixuanq@andrew.cmu.edu

**Lingsong Zhang & Xiao Wang**
Department of Statistics
Purdue University
West Lafayette, IN 47907, USA
{lingsong, wangxiao}@purdue.edu

## Abstract

The contrastive divergence algorithm is a popular approach to training energy-based latent variable models, which has been widely used in many machine learning models such as the restricted Boltzmann machines and deep belief nets. Despite its empirical success, the contrastive divergence algorithm is also known to have biases that severely affect its convergence. In this article we propose an unbiased version of the contrastive divergence algorithm that completely removes its bias in stochastic gradient methods, based on recent advances on unbiased Markov chain Monte Carlo methods. Rigorous theoretical analysis is developed to justify the proposed algorithm, and numerical experiments show that it significantly improves the existing method. Our findings suggest that the unbiased contrastive divergence algorithm is a promising approach to training general energy-based latent variable models.

## 1 Introduction

Energy-based latent variable models cover a broad class of generative models that are frequently used to characterize sophisticated distributions of high-dimensional data. Popular examples of this class include the restricted Boltzmann machines (RBM, Smolensky, 1986; Hinton, 2012), deep belief nets (Hinton et al., 2006), and exponential family harmoniums (Welling et al., 2005), among many others. Energy-based models are complementary to directed generative models such as the variational autoencoders (Kingma & Welling, 2014), and have gained great success in synthesizing realistic data samples (Xie et al., 2016; 2018b). They can also be combined with directed models to build more sophisticated structures (Xie et al., 2018a). In this article we focus on the energy-based latent variable model, whose general form can be expressed in terms of the joint distribution of a visible random vector, $\mathbf{v} \in \mathbb{V} \subset \mathbb{R}^p$, and a hidden or latent random vector, $\mathbf{h} \in \mathbb{H} \subset \mathbb{R}^r$, with the density function

$$p(\boldsymbol{v}, \boldsymbol{h}; \boldsymbol{\theta}) = \frac{1}{Z(\boldsymbol{\theta})} \exp\{-E(\boldsymbol{v}, \boldsymbol{h}; \boldsymbol{\theta})\}, \tag{1}$$

where $\boldsymbol{\theta} \in \Theta$ is the unknown parameter vector, $E(\boldsymbol{v}, \boldsymbol{h}; \boldsymbol{\theta})$ is the energy function, and $Z(\boldsymbol{\theta})$ is a normalizing constant to ensure that $p(\boldsymbol{v}, \boldsymbol{h}; \boldsymbol{\theta})$ is a legitimate probability density or mass function. The model distribution, $p_{\mathbf{v}}(\boldsymbol{v}; \boldsymbol{\theta})$, is defined to be the marginal distribution of $p(\boldsymbol{v}, \boldsymbol{h}; \boldsymbol{\theta})$.

Similar to many other machine learning models, the standard approach to estimating the parameter vector $\boldsymbol{\theta}$ is the maximum likelihood method. It can be shown that the derivative of the log-likelihood function can be expressed as the difference of two expectations, and hence Monte Carlo methods, especially the Markov chain Monte Carlo (MCMC, Gilks et al., 1995), can be used to approximate the gradient. Various optimization techniques, such as the stochastic gradient method (SG, Robbins & Monro, 1951; Bottou, 2010), can then proceed to iteratively update the parameter estimate. This strategy, though elegant in theory, is not without limitations. In particular, MCMC estimators are typically consistent in the limiting case, but biased on finite steps, so one needs to run MCMC for a long time to obtain an accurate gradient, which would take tremendous amount of computing time.

To reduce the computational complexity, Hinton (2002) proposed a simple and fast algorithm, called the contrastive divergence (CD) algorithm. The basic idea of CD is to truncate MCMC at the $k$-th step, and use the resulting approximate gradient to update $\boldsymbol{\theta}$, where $k$ is a fixed integer as small as one. Such an approach is usually referred to as the CD-$k$ algorithm. The simplicity and computational efficiency of CD makes it widely used in many popular energy-based models, and there was also numerous empirical evidence to illustrate the effectiveness of CD. More recently, the CD idea is applied to directed generative models (Ruiz & Titsias, 2019), where CD is used to define a loss function that combines variational inference and MCMC.

However, the success of CD also raised a lot of questions regarding its convergence properties. Both theoretical and empirical results show that CD in general does not converge to a local minimum of the likelihood function (Carreira-Perpiñán & Hinton, 2005), and diverges even in some simple models (Schulz et al., 2010; Fischer & Igel, 2010). The main issue of CD is that the truncation of MCMC produces a biased stochastic gradient for the log-likelihood function in every iteration, and such uncontrolled biases may be accumulated to distort the true ascent direction. Due to this reason, the training of energy-based models has been a longstanding challenge in machine learning research.

In this article, we propose a new unbiased contrastive divergence (UCD) algorithm based on recent advances in unbiased MCMC theory, which offers new possibilities for solving the model training problem. In the seminal work Glynn & Rhee (2014), the authors developed an unbiased estimator for the expectation with respect to the invariant distribution of a Markov chain. More recently, this estimator was further extended to the MCMC setting by Jacob et al. (2017), using a technique called coupling. At a high level, by carefully designing the MCMC algorithm, one is able to get an unbiased MCMC estimator with only finite number of Markov transitions.

Under the framework of Glynn & Rhee (2014) and Jacob et al. (2017), the proposed UCD algorithm is a Gibbs-sampler-based training method for energy-based latent variable models. We prove that the stochastic gradient generated by UCD is unbiased with a finite variance, which implies the convergence of SG based on it. Similar to CD-$k$, UCD generates Markov chains to compute the gradient, but the chain in UCD stops at a random time, instead of a fixed one as in CD-$k$. The theoretical analysis indicates that the stopping time has a finite expectation, so on average the computation can be completed in finite time. Besides theoretical justifications, our numerical experiments show that UCD significantly improves existing training algorithms, suggesting that it is a promising approach with a solid convergence guarantee. The implementation of the UCD algorithm is available at `https://github.com/yixuan/cdtau`. The highlights of this article are as follows:

- We develop a new training algorithm for general energy-based latent variable models that include many popular models (e.g. RBM) as special cases. To our best knowledge, this is the first algorithm that has a solid convergence guarantee for such models.

- The proposed algorithm resolves a longstanding problem of the CD algorithm, the bias in approximating the gradient. In particular, our method is completely unbiased, and theoretical justifications are developed to guarantee its convergence.

- We have tailored a specialized algorithm for RBM, which is shown to significantly reduce the computational cost.

## 2  A BRIEF REVIEW OF CONTRASTIVE DIVERGENCE

In this section we briefly review the CD algorithm, and point out some of its weaknesses that have been studied in the existing literature. For a single observation $\boldsymbol{v}$, the marginal data log-likelihood function is $\ell(\boldsymbol{\theta}; \boldsymbol{v}) = \log\{p_{\mathbf{v}}(\boldsymbol{v}; \boldsymbol{\theta})\} = \log\{\int p(\boldsymbol{v}, \boldsymbol{h}; \boldsymbol{\theta})\mathrm{d}\boldsymbol{h}\}$. Assume that $E(\boldsymbol{v}, \boldsymbol{h}; \boldsymbol{\theta})$ is continuously differentiable for $\boldsymbol{\theta}$, and then with $n$ data points $\mathcal{D} = (\mathbf{v}_1, \ldots, \mathbf{v}_n)$, the derivative of the log-likelihood function $\ell(\boldsymbol{\theta}; \mathcal{D}) = \sum \ell(\boldsymbol{\theta}; \mathbf{v}_i)$, also known as the score function, can be written as

$$\frac{\partial \ell(\boldsymbol{\theta}; \mathcal{D})}{\partial \boldsymbol{\theta}} = -n\left[\mathbb{E}_{(\mathbf{v},\mathbf{h}) \sim p(\mathcal{D})p(\boldsymbol{h}|\boldsymbol{v};\boldsymbol{\theta})}\left\{\frac{\partial E(\mathbf{v}, \mathbf{h}; \boldsymbol{\theta})}{\partial \boldsymbol{\theta}}\right\} - \mathbb{E}_{(\mathbf{v},\mathbf{h}) \sim p(\boldsymbol{v},\boldsymbol{h};\boldsymbol{\theta})}\left\{\frac{\partial E(\mathbf{v}, \mathbf{h}; \boldsymbol{\theta})}{\partial \boldsymbol{\theta}}\right\}\right], \quad (2)$$

where $p(\mathcal{D})$ stands for the empirical distribution of $\mathcal{D}$, and $p(\boldsymbol{h}|\boldsymbol{v}; \boldsymbol{\theta})$ is the conditional distribution of the latent variable $\mathbf{h}$ given $\mathbf{v} = \boldsymbol{v}$. A simple derivation of (2) can be found in Fischer & Igel (2014). Throughout this article we denote $\mathbf{x} = (\mathbf{v}, \mathbf{h}) \in \mathbb{X} := \mathbb{V} \times \mathbb{H}$ and $f(\boldsymbol{x}; \boldsymbol{\theta}) = \partial E(\boldsymbol{v}, \boldsymbol{h}; \boldsymbol{\theta})/\partial \boldsymbol{\theta}$. Then

the two expectations in (2) can be abbreviated as $\mathbb{E}_{\mathcal{D}}\{f(\mathbf{x};\boldsymbol{\theta})\}$ and $\mathbb{E}_{\mathcal{M}}\{f(\mathbf{x};\boldsymbol{\theta})\}$, respectively, where $\mathcal{M} := p(\boldsymbol{v},\boldsymbol{h};\boldsymbol{\theta})$ is the complete model distribution.

In many cases, for example the RBM model, $\mathbb{E}_{\mathcal{D}}\{f(\mathbf{x};\boldsymbol{\theta})\}$ has a closed form, so the major computational difficulty comes from the $\mathbb{E}_{\mathcal{M}}\{f(\mathbf{x};\boldsymbol{\theta})\}$ term. A common scheme to approximate this expectation is to run a Markov chain $\xi_0 \to \xi_1 \to \cdots$ with $\mathcal{M}$ as the invariant distribution, and then under mild conditions we have $\lim_{t\to\infty}\mathbb{E}\{f(\xi_t;\boldsymbol{\theta})\} = \mathbb{E}_{\mathcal{M}}\{f(\mathbf{x};\boldsymbol{\theta})\}$. Of course, such a limit cannot be reached in finite steps, so the CD-$k$ algorithm truncates the Markov chain at the $k$-th step, resulting in the following approximation:

$$\Delta(\boldsymbol{\theta}) := -\left[\mathbb{E}_{\mathcal{D}}\{f(\mathbf{x};\boldsymbol{\theta})\} - f(\xi_k;\boldsymbol{\theta})\right]. \tag{3}$$

It is easy to see that $\Delta(\boldsymbol{\theta}) \approx n^{-1}\partial\ell(\boldsymbol{\theta};\mathcal{D})/\partial\boldsymbol{\theta}$ is a stochastic approximation to the true gradient, so one can use SG to update $\boldsymbol{\theta}$ via the iteration $\boldsymbol{\theta}_{i+1} = \boldsymbol{\theta}_i + \alpha_i\Delta(\boldsymbol{\theta}_i)$, where $\boldsymbol{\theta}_i$ is the parameter estimate in the $i$-th iteration, and $\alpha_i$ is the step size.

Despite its simplicity, various research articles have pointed out the weaknesses of the CD-$k$ algorithm. For instance, Sutskever & Tieleman (2010) gave an example to show that $\mathbb{E}\{\Delta(\boldsymbol{\theta})\}$ is not the gradient of any objective function, and Schulz et al. (2010); Fischer & Igel (2014) studied numerical experiments in which CD-$k$ does not converge at all for small $k$ values. Carreira-Perpiñán & Hinton (2005) considered the fixed points of $\Delta(\boldsymbol{\theta})$, the $\boldsymbol{\theta}$ values such that $\mathbb{E}\{\Delta(\boldsymbol{\theta})\} = 0$, and showed that they do not match the fixed points of $\partial\ell(\boldsymbol{\theta};\mathcal{D})/\partial\boldsymbol{\theta}$ in general. This implies that even if CD-$k$ converges, the resulting parameter estimate may not be a local minimum of the likelihood function.

Another variant of CD is the persistent contrastive divergence (PCD, Tieleman, 2008; Tieleman & Hinton, 2009), which has been reported to improve CD in many numerical experiments. However, it is still an approximation method, and its convergence properties are more difficult to analyze, as the stochastic gradients generated by PCD become correlated across iterations. In fact, Schulz et al. (2010); Fischer & Igel (2010) also gave examples in which PCD failed to converge. There are also some other training methods as extensions to CD, such as the multi-grid method (Gao et al., 2018) and the short-run MCMC (Nijkamp et al., 2019), but all these methods inherit the bias of CD.

To summarize, it is surprising that virtually none of the popular training methods for energy-based models, including CD and PCD, provide a solid convergence guarantee. The major defects of CD stem from the fact that $\Delta(\boldsymbol{\theta})$ is a biased estimator for the true log-likelihood gradient, and SG may fail with uncontrolled bias accumulation. To this end, the ultimate solution is to design a training algorithm that completely removes the bias of CD.

## 3 THE UNBIASED CONTRASTIVE DIVERGENCE ALGORITHM

### 3.1 UNBIASED MCMC ESTIMATORS

Since CD highly relies on the MCMC method, the main ingredient of the proposed UCD algorithm is the theory of unbiased MCMC developed by Glynn & Rhee (2014) and Jacob et al. (2017). Consider the second term in (2), namely, $\mathbb{E}_{\mathcal{M}}\{f(\mathbf{x};\boldsymbol{\theta})\}$. In what follows we omit the dependence on $\boldsymbol{\theta}$ for brevity if no confusion is caused. If a Markov chain $\{\xi_t\}$ satisfies $\mathbb{E}\{f(\xi_t)\} \to \mathbb{E}_{\mathcal{M}}\{f(\mathbf{x})\}$ as $t \to \infty$, then under some regularity conditions, we can express the limit as a telescoping sum,

$$\mathbb{E}_{\mathcal{M}}\{f(\mathbf{x})\} = \mathbb{E}\{f(\xi_k)\} + \sum_{t=k+1}^{\infty}\left[\mathbb{E}\{f(\xi_t)\} - \mathbb{E}\{f(\xi_{t-1})\}\right]$$

for any fixed $k \geq 0$. Now assume that there exists another Markov chain $\{\eta_t\}$ such that $\xi_t$ and $\eta_t$ have the same marginal distributions for all $t \geq 0$, and $\xi_t = \eta_{t-1}$ for all $t \geq \tau$, where $\tau$ is some random time. If we allow the exchange of expectation and summation, then we would get

$$\mathbb{E}_{\mathcal{M}}\{f(\mathbf{x})\} = \mathbb{E}\left[f(\xi_k) + \sum_{t=k+1}^{\infty}\{f(\xi_t) - f(\eta_{t-1})\}\right] = \mathbb{E}\left[f(\xi_k) + \sum_{t=k+1}^{\tau-1}\{f(\xi_t) - f(\eta_{t-1})\}\right],$$

where the first identity holds since $\mathbb{E}\{f(\xi_t)\} = \mathbb{E}\{f(\eta_t)\}$ for all $t \geq 0$, and the second one is due to the fact that $\xi_t = \eta_{t-1}$ for $t \geq \tau$. As a consequence, the quantity $f(\xi_k) + \sum_{t=k+1}^{\tau-1}\{f(\xi_t) - f(\eta_{t-1})\}$ is an unbiased estimator for $\mathbb{E}_{\mathcal{M}}\{f(\mathbf{x})\}$. Such an idea seems rather simple, but the construction of the chain $\{\eta_t\}$, which we describe in the next section, is a highly non-trivial task.

### 3.2 COUPLING OF MARKOV CHAINS

Let $\mathcal{M}_t$ denote the marginal distribution of a Markov chain $\{\xi_t\}$ at the $t$-th step. By construction, $\mathcal{M}_t$ converges to $\mathcal{M}$ as $t \to \infty$. To develop the unbiased estimator $H_k(\xi, \eta)$, the second chain $\{\eta_t\}$ must satisfy two conditions: (1) marginally $\eta_t \sim \mathcal{M}_t$; (2) $\{\xi_t\}$ and the lag-one sequence $\{\eta_{t-1}\}$ will meet and stay identical after some random time $\tau$. Condition (1) can be trivially met if $\{\xi_t\}$ and $\{\eta_t\}$ are sampled independently. However, in this way the probability that $\xi_t = \eta_{t-1}$ may be extremely small, or even be zero for continuous random variables. Therefore, a special joint distribution for $(\xi_t, \eta_{t-1})$ needs to be assigned subject to $\xi_t \sim \mathcal{M}_t$ and $\eta_{t-1} \sim \mathcal{M}_{t-1}$. Such a pair of random variables under the marginal distribution constraints is called a *coupling*, and for our purpose we attempt to seek a coupling scheme such that $P(\xi_t = \eta_{t-1}) > 0$. Figure 1 illustrates the coupling process of two Markov chains $\{\xi_t\}$ and $\{\eta_{t-1}\}$.

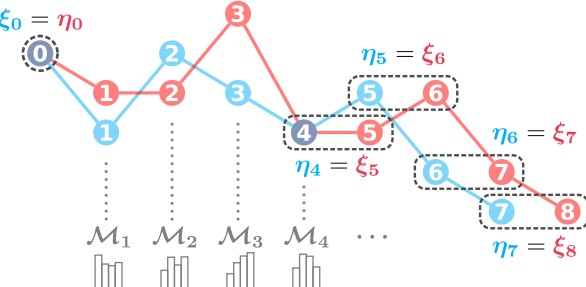

Figure 1: An illustration of the coupling process. $\{\xi_t\}$ and $\{\eta_t\}$ start from the same value, and have the same marginal distribution $\mathcal{M}_t$ at each step. The two chains are correlated in such a way that the event $\xi_t = \eta_{t-1}$ occurs with a positive probability for each $t$. After a random time $\tau$ ($\tau = 5$ in the illustration), $\{\xi_t\}$ meets $\{\eta_{t-1}\}$ and they stay identical afterwards.

To implement such a coupling, first let $\{\xi_t\}$ and $\{\eta_t\}$ start from the same initial value $\xi_0 = \eta_0$, and additionally draw $\xi_1 \sim \mathcal{T}(\cdot|\xi_0)$, where $\mathcal{T}(\boldsymbol{y}|\boldsymbol{x})$ stands for the transition density function from state $\boldsymbol{x}$ to state $\boldsymbol{y}$. Next, we need to draw $(\xi_2, \eta_1)$ such that marginally $\xi_2 \sim \mathcal{M}_2$ and $\eta_1 \sim \mathcal{M}_1$, which can be a difficult task as $\mathcal{M}_t$ may not have closed forms. Fortunately, it is much simplified for Markov chains: due to the Markov property, $\xi_2$ and $\eta_1$ will have the requested marginal distributions if we sample $\xi_2|\xi_1 \sim \mathcal{T}(\cdot|\xi_1)$ and $\eta_1|\eta_0 \sim \mathcal{T}(\cdot|\eta_0)$ conditional on $\xi_1$ and $\eta_0$. That is, the coupling of Markov chains can be achieved by the coupling of one-step transitions, which is a much simpler task. Define two density functions $p(\cdot) = \mathcal{T}(\cdot|\xi_1)$ and $q(\cdot) = \mathcal{T}(\cdot|\eta_0)$, and then the problem reduces to drawing a coupling $(\xi, \eta)$ such that $\xi \sim p(\cdot)$, $\eta \sim q(\cdot)$, and $P(\xi = \eta) > 0$, which can be accomplished via the maximal coupling technique (Appendix A.1).

Specific to our problem (2), we need to sample $\mathbf{x} = (\mathbf{v}, \mathbf{h})$ from $p(\boldsymbol{v}, \boldsymbol{h}; \boldsymbol{\theta})$. In energy-based latent variable models, the most widely-used MCMC method is the Gibbs sampler (Geman & Geman, 1984), which sequentially updates one block of $\mathbf{x}$ based on the conditional distribution of this block given the rest. As an example, in RBM models $\mathbf{v}|\{\mathbf{h} = \boldsymbol{h}\}$ and $\mathbf{h}|\{\mathbf{v} = \boldsymbol{v}\}$ follow multivariate Bernoulli distributions with independent components, which are very easy to sample from. The coupling for Gibbs samplers was briefly mentioned in Jacob et al. (2017) as a special case of the Metropolis–Hastings scheme (Metropolis et al., 1953; Hastings, 1970), but next we show that some specific structure of Gibbs samplers can be utilized to simplify the process.

For simplicity and clarity, we assume that the Gibbs sampler for $\mathcal{M}$ follow the natural division of blocks $\mathbf{x} = (\mathbf{v}, \mathbf{h})$. That is, one can easily sample from the two transition distributions $\mathcal{T}_v(\boldsymbol{v}|\boldsymbol{h}) := p(\boldsymbol{v}|\boldsymbol{h}; \boldsymbol{\theta})$ and $\mathcal{T}_h(\boldsymbol{h}|\boldsymbol{v}) := p(\boldsymbol{h}|\boldsymbol{v}; \boldsymbol{\theta})$. The more sophisticated cases, for example $\mathbf{h}$ consists of multiple layers $\mathbf{h} = (\mathbf{h}_1, \ldots, \mathbf{h}_L)$, can be dealt with similarly. In Algorithm 1, we describe the steps to sample two coupled chains $\{\xi_t = (\boldsymbol{v}_t, \boldsymbol{h}_t)\}$ and $\{\eta_t = (\boldsymbol{v}'_t, \boldsymbol{h}'_t)\}$ based on the Gibbs sampler.

Three remarks are made for Algorithm 1: (1) The meeting event (line 4) only depends on the $\mathcal{T}_v$ transition density. To verify this, note that at the $t$-th step, we need to draw $\xi_t|\xi_{t-1} \sim \mathcal{T}(\cdot|\xi_{t-1})$ and $\eta_{t-1}|\eta_{t-2} \sim \mathcal{T}(\cdot|\eta_{t-2})$, where $\mathcal{T}(\tilde{\boldsymbol{v}}, \tilde{\boldsymbol{h}}|\boldsymbol{v}, \boldsymbol{h}) = \mathcal{T}_v(\tilde{\boldsymbol{v}}|\boldsymbol{h})\mathcal{T}_h(\tilde{\boldsymbol{h}}|\tilde{\boldsymbol{v}})$ is the transition density for a full update cycle. It is easy to show that $\mathcal{T}(\tilde{\boldsymbol{v}}, \tilde{\boldsymbol{h}}|\boldsymbol{v}', \boldsymbol{h}')/\mathcal{T}(\tilde{\boldsymbol{v}}, \tilde{\boldsymbol{h}}|\boldsymbol{v}, \boldsymbol{h}) = \mathcal{T}_v(\tilde{\boldsymbol{v}}|\boldsymbol{h}')/\mathcal{T}_v(\tilde{\boldsymbol{v}}|\boldsymbol{h})$, so the $\mathcal{T}_h$ part cancels in the ratio. (2) Once $\xi_t$ and $\eta_{t-1}$ meet, they stay identical afterwards, because by then $\mathcal{T}_v(\cdot|\boldsymbol{h}'_{t-2}) = \mathcal{T}_v(\cdot|\boldsymbol{h}_{t-1})$, and the event in line 4 always happens. (3) Line 7 is a rejection sampling step. In our numerical experiments we find that very few samples are rejected, so its cost is tiny.

---

**Algorithm 1** Coupling method for the Gibbs sampler

---

**Input:** Densities $\mathcal{T}_v(\boldsymbol{v}|\boldsymbol{h})$ and $\mathcal{T}_h(\boldsymbol{h}|\boldsymbol{v})$, initial values $\xi_0 = (\boldsymbol{v}_0, \boldsymbol{h}_0) = \eta_0 = (\boldsymbol{v}_0', \boldsymbol{h}_0')$, $T_{\max}$
**Output:** Coupled chains $\{\xi_t\}$ and $\{\eta_t\}$
  1: Sample $\boldsymbol{v}_1 \sim \mathcal{T}_v(\cdot|\boldsymbol{h}_0)$ and $\boldsymbol{h}_1 \sim \mathcal{T}_h(\cdot|\boldsymbol{v}_1)$. Set $\xi_1 = (\boldsymbol{v}_1, \boldsymbol{h}_1)$
  2: **for** $t = 2, 3, \ldots$ **do**
  3:    Sample $\boldsymbol{v}_t \sim \mathcal{T}_v(\cdot|\boldsymbol{h}_{t-1})$, $\boldsymbol{h}_t \sim \mathcal{T}_h(\cdot|\boldsymbol{v}_t)$, and $U \sim \mathsf{Uniform}(0, 1)$
  4:    **if** $U \leq \mathcal{T}_v(\boldsymbol{v}_t|\boldsymbol{h}_{t-2}')/\mathcal{T}_v(\boldsymbol{v}_t|\boldsymbol{h}_{t-1})$ or $t \geq T_{\max}$ (maximum stopping time) **then**
  5:       Set $\xi_t = (\boldsymbol{v}_t, \boldsymbol{h}_t)$, $\eta_{t-1} = \xi_t$
  6:    **else**
  7:       Sample $\boldsymbol{v}_{t-1}' \sim \mathcal{T}_v(\cdot|\boldsymbol{h}_{t-2}')$, $\boldsymbol{h}_{t-1}' \sim \mathcal{T}_h(\cdot|\boldsymbol{v}_{t-1}')$, and $U' \sim \mathsf{Uniform}(0, 1)$
         until $U' > \mathcal{T}_v(\boldsymbol{v}_{t-1}'|\boldsymbol{h}_{t-1})/\mathcal{T}_v(\boldsymbol{v}_{t-1}'|\boldsymbol{h}_{t-2}')$
  8:       Set $\xi_t = (\boldsymbol{v}_t, \boldsymbol{h}_t)$, $\eta_{t-1} = (\boldsymbol{v}_{t-1}', \boldsymbol{h}_{t-1}')$
  9:    **end if**
10: **end for**

---

### 3.3 Unbiased Contrastive Divergence

The technical tools introduced in Sections 3.1 and 3.2 enable us to develop a new algorithm to train model (1). Recall that the true gradient of the log-likelihood function is given by (2). The first term, $\mathbb{E}_{\mathcal{D}}\{f(\mathbf{x}; \boldsymbol{\theta})\}$, can be computed exactly, and the second term, $\mathbb{E}_{\mathcal{M}}\{f(\mathbf{x}; \boldsymbol{\theta})\}$, is approximated by an unbiased estimator $\tilde{g}_2(\boldsymbol{\theta}) := f(\xi_k) + \sum_{t=k+1}^{\tau-1}\{f(\xi_t) - f(\eta_{t-1})\}$, where the coupled Markov chains $\{\xi_t\}$ and $\{\eta_t\}$ are generated by Algorithm 1. Assume that the parameter vector $\boldsymbol{\theta}$ lies in a closed convex set $\Theta$, and let $\mathcal{P}_\Theta(\cdot)$ denote the projection onto $\Theta$. Putting the pieces together, Algorithm 2 illustrates the UCD algorithm for training energy-based latent variable models. The initial chain length $k$ can be any fixed number, and in this article we take $k = 1$ for all the numerical experiments.

---

**Algorithm 2** UCD Algorithm for estimating $\boldsymbol{\theta}$

---

**Input:** $T$, $\{\alpha_i\}$, $k$, initial value $\boldsymbol{\theta}_0$
**Output:** Parameter estimate for $\boldsymbol{\theta}$
  1: **for** $i = 0, 1, \ldots, T - 1$ **do**
  2:    Draw one data point $\boldsymbol{v} \sim p(\mathcal{D})$, and sample $\boldsymbol{h} \sim p(\boldsymbol{h}|\boldsymbol{v}; \boldsymbol{\theta}_i)$
  3:    Set $\xi_0 = \eta_0 = (\boldsymbol{v}, \boldsymbol{h})$, and run Algorithm 1 with $\boldsymbol{\theta} = \boldsymbol{\theta}_i$ until $\xi_{\tau_i} = \eta_{\tau_i - 1}$
  4:    $\tilde{g}(\boldsymbol{\theta}) \leftarrow -\mathbb{E}_{\mathbf{h} \sim p(\boldsymbol{h}|\boldsymbol{v}; \boldsymbol{\theta})}\{f(\boldsymbol{v}, \mathbf{h}; \boldsymbol{\theta})\} + f(\xi_k) + \sum_{t=k+1}^{\tau_i - 1}\{f(\xi_t) - f(\eta_{t-1})\}$
  5:    $\boldsymbol{\theta}_{i+1} \leftarrow \mathcal{P}_\Theta\left(\boldsymbol{\theta}_i + \alpha_i \cdot \tilde{g}(\boldsymbol{\theta}_i)\right)$
  6: **end for**
  7: **return** $\hat{\boldsymbol{\theta}} = T^{-1}\sum_{i=1}^{T}\boldsymbol{\theta}_i$

---

Next, we analyze the theoretical property of Algorithm 2 and state the conditions for it to converge. As a standard setting, we assume that the Markov chains generated by the Gibbs sampler are $\varphi$-irreducible and aperiodic (Meyn & Tweedie, 2012). This is a very mild assumption that every practical Gibbs sampler should satisfy. Then we make the following two assumptions that guarantee the convergence of Gibbs samplers.

**Assumption 1.** *(Drift condition) There exist a pair of functions $r : \mathbb{V} \to [1, +\infty)$, $l : \mathbb{H} \to [1, +\infty)$ and constants $\gamma_1, \gamma_2, L_1, L_2 > 0$ such that $\gamma_1\gamma_2 < 1$ and*

$$\mathbb{E}_{\mathbf{v} \sim p(\boldsymbol{v}|\boldsymbol{h}; \boldsymbol{\theta})}r(\mathbf{v}) \leq \gamma_1 l(\boldsymbol{h}) + L_1, \quad \mathbb{E}_{\mathbf{h} \sim p(\boldsymbol{h}|\boldsymbol{v}; \boldsymbol{\theta})}l(\mathbf{h}) \leq \gamma_2 r(\boldsymbol{v}) + L_2, \quad \forall \boldsymbol{v} \in \mathbb{V}, \ \boldsymbol{h} \in \mathbb{H}, \ \boldsymbol{\theta} \in \Theta.$$

*Also, there exist constants $c > 0$ and $D > 0$ such that $|f(\boldsymbol{x}; \boldsymbol{\theta})|^{2+c} \leq l(\boldsymbol{h})$ and $\mathbb{E}_{\mathbf{h} \sim p_{\mathbf{h}}(\boldsymbol{h}; \boldsymbol{\theta})}l(\mathbf{h}) \leq D$ for all $\boldsymbol{x} = (\boldsymbol{v}, \boldsymbol{h}) \in \mathbb{X}$ and $\boldsymbol{\theta} \in \Theta$.*

**Assumption 2.** *(Minorization condition) There exist constants $d > 2(\gamma_2 L_1 + L_2)/(1 - \gamma_1\gamma_2)$, $\varepsilon > 0$, and a density function $q(\cdot)$ such that $p(\boldsymbol{v}|\boldsymbol{h}; \boldsymbol{\theta}) \geq \varepsilon q(\boldsymbol{v})$ for all $\boldsymbol{h} \in \mathbb{D}$, $\boldsymbol{v} \in \mathbb{V}$, and $\boldsymbol{\theta} \in \Theta$, where $\mathbb{D} = \{\boldsymbol{h} \in \mathbb{H} : l(\boldsymbol{h}) \leq d\}$.*

In the following theorem we show three important facts about the proposed stochastic gradient $\tilde{g}(\theta)$: (1) $\tilde{g}(\theta)$ is unbiased for the true score function; (2) it has a bounded second moment uniformly in $\boldsymbol{\theta}$; (3) in expectation it can be computed in finite time.

**Theorem 1.** *Under Assumptions 1 and 2, there exist constants $D_1, D_2 > 0$ such that $\mathbb{E}\{\tilde{g}(\boldsymbol{\theta})\} = \partial\ell(\boldsymbol{\theta}; \boldsymbol{v})/\partial\boldsymbol{\theta}$, $\mathbb{E}\left[\{\tilde{g}_2(\boldsymbol{\theta})\}^2\right] \leq D_1$, and $\mathbb{E}(\tau_i) \leq D_2$ for all $\boldsymbol{\theta} \in \Theta$ and $i = 1, 2, \ldots, T-1$.*

Theorem 1 provides the building blocks for the convergence analysis of Algorithm 2. With the unbiased gradient estimator and the bounded second moment, we establish a solid convergence guarantee for the proposed algorithm. As a typical setting, in the following corollary we consider a convex log-likelihood function.

**Corollary 1.** *Assume that $\ell(\boldsymbol{\theta}; \boldsymbol{v})$ is convex and $L$-Lipschitz continuous in $\boldsymbol{\theta} \in \Theta$, and $\Theta$ is a closed and bounded convex set. Then by choosing $\alpha_i = \alpha_0/\sqrt{i}$ for some constant $\alpha_0 > 0$, we have $\ell^* - \ell(\hat{\boldsymbol{\theta}}; \boldsymbol{v}) \leq \mathcal{O}(1/\sqrt{T})$, where $\ell^*$ is the maximum value of $\ell(\boldsymbol{\theta}; \boldsymbol{v})$.*

The proof Corollary 1 is standard, see for example Bottou (2010); Bottou et al. (2018). When the log-likelihood function is nonconvex as in the RBM model, there are also other versions of the convergence result for SG, for example Theorem 4.10 of Bottou et al. (2018). Such directions can be studied separately and are omitted here.

Finally, we shall point out an important special case of Theorem 1, *i.e.*, if the Markov chain $\{\xi_t\}$ has finite states, then the two assumptions are automatically satisfied. This shows that many widely-used models, for example RBM, can directly use the UCD algorithm without the need to find such $r(\cdot)$ and $l(\cdot)$ functions. We summarize this useful fact in the following corollary.

**Corollary 2.** *If $\mathbb{X}$ is a finite state space and $\Theta$ is compact, then Assumptions 1 and 2 hold, and Theorem 1 applies.*

## 4 TRAINING RESTRICTED BOLTZMANN MACHINES

RBM is one of the most popular and widely-used energy models in machine learning, defined by the energy function $E(\boldsymbol{v}, \boldsymbol{h}; \boldsymbol{\theta}) = -\boldsymbol{v}^{\mathrm{T}}\boldsymbol{b} - \boldsymbol{v}^{\mathrm{T}}\boldsymbol{W}\boldsymbol{h} - \boldsymbol{h}^{\mathrm{T}}\boldsymbol{c}$, where $\boldsymbol{v} \in \{0, 1\}^m$, $\boldsymbol{h} \in \{0, 1\}^n$, and $\boldsymbol{\theta} = (\boldsymbol{W}, \boldsymbol{b}, \boldsymbol{c})$ are model parameters. The Gibbs sampler for RBM has a nice structure: let $\sigma(\boldsymbol{x}) = 1/(1 + \exp(-\boldsymbol{x}))$ be the sigmoid function, and then $\mathbf{v}|\{\mathbf{h} = \boldsymbol{h}\} \sim \mathsf{Bernoulli}(\sigma(\boldsymbol{W}\boldsymbol{h} + \boldsymbol{b}))$ and $\mathbf{h}|\{\mathbf{v} = \boldsymbol{v}\} \sim \mathsf{Bernoulli}(\sigma(\boldsymbol{W}^{\mathrm{T}}\boldsymbol{v} + \boldsymbol{c}))$. The coupling method in Algorithm 1 directly works for RBM, but here we show an improved version that is tailored for RBM and is more efficient.

Let $\boldsymbol{u}, \boldsymbol{p} \in \mathbb{R}^r$, and the notation $\boldsymbol{y} = \mathbf{1}\{\boldsymbol{u} \leq \boldsymbol{p}\}$ stands for a binary vector such that $y_i = 1$ if $u_i \leq p_i$ and $y_i = 0$ otherwise. Also let $\mathcal{T}_v(\boldsymbol{v}|\boldsymbol{h}) = \prod_{i=1}^{m} p_i^{v_i}(1 - p_i)^{1-v_i}$ denote the transition density from $\boldsymbol{h}$ to $\boldsymbol{v}$, where $\boldsymbol{p} = (p_1, \ldots, p_m)^{\mathrm{T}} = \sigma(\boldsymbol{W}\boldsymbol{h} + \boldsymbol{b})$. Then the specialized coupling method for RBM is given in Algorithm 3.

---

**Algorithm 3** Coupling method for RBM

---

**Input:** Model parameters $\boldsymbol{W}, \boldsymbol{b}, \boldsymbol{c}$, step-$t$ states $\xi_t = (\boldsymbol{v}_t, \boldsymbol{h}_t)$, $\eta_{t-1} = (\boldsymbol{v}'_{t-1}, \boldsymbol{h}'_{t-1})$
**Output:** New states $\xi_{t+1} = (\boldsymbol{v}_{t+1}, \boldsymbol{h}_{t+1})$, $\eta_t = (\boldsymbol{v}'_t, \boldsymbol{h}'_t)$
1: Sample $U_1 \sim \mathsf{Uniform}(0, 1)$, $\boldsymbol{Z}_1 \sim \mathsf{Uniform}([0, 1]^m)$, and set $\boldsymbol{v}_{t+1} = \mathbf{1}\{\boldsymbol{Z}_1 \leq \sigma(\boldsymbol{W}\boldsymbol{h}_t + \boldsymbol{b})\}$
2: **if** $U_1 \leq \mathcal{T}_v(\boldsymbol{v}_{t+1}|\boldsymbol{h}'_{t-1})/\mathcal{T}_v(\boldsymbol{v}_{t+1}|\boldsymbol{h}_t)$ **then**
3:     Set $\boldsymbol{v}'_t = \boldsymbol{v}_{t+1}$
4: **else**
5:     **repeat**
6:         Sample $U_2 \sim \mathsf{Uniform}(0, 1)$, $U'_2 \sim \mathsf{Uniform}(0, 1)$, $\boldsymbol{Z}_2 \sim \mathsf{Uniform}([0, 1]^m)$
7:         **if** $\boldsymbol{v}_{t+1}$ has not been accepted **then**
8:             Propose $\boldsymbol{v}_{t+1} = \mathbf{1}\{\boldsymbol{Z}_2 \leq \sigma(\boldsymbol{W}\boldsymbol{h}_t + \boldsymbol{b})\}$, accept if $U_2 > \mathcal{T}_v(\boldsymbol{v}_{t+1}|\boldsymbol{h}'_{t-1})/\mathcal{T}_v(\boldsymbol{v}_{t+1}|\boldsymbol{h}_t)$
9:         **end if**
10:        **if** $\boldsymbol{v}'_t$ has not been accepted **then**
11:           Propose $\boldsymbol{v}'_t = \mathbf{1}\{\boldsymbol{Z}_2 \leq \sigma(\boldsymbol{W}\boldsymbol{h}'_{t-1} + \boldsymbol{b})\}$, accept if $U'_2 > \mathcal{T}_v(\boldsymbol{v}'_t|\boldsymbol{h}_t)/\mathcal{T}_v(\boldsymbol{v}'_t|\boldsymbol{h}'_{t-1})$
12:        **end if**
13:     **until** $\boldsymbol{v}_{t+1}$ and $\boldsymbol{v}'_t$ are both accepted
14: **end if**
15: Sample $\boldsymbol{Z}_3 \sim \mathsf{Uniform}([0, 1]^n)$
16: Set $\boldsymbol{h}_{t+1} = \mathbf{1}\{\boldsymbol{Z}_3 \leq \sigma(\boldsymbol{W}^{\mathrm{T}}\boldsymbol{v}_{t+1} + \boldsymbol{c})\}$, $\boldsymbol{h}'_t = \mathbf{1}\{\boldsymbol{Z}_3 \leq \sigma(\boldsymbol{W}^{\mathrm{T}}\boldsymbol{v}'_t + \boldsymbol{c})\}$

---

The intuition behind Algorithm 3 is the following: line 2 indicates that it is also a maximal coupling method, so the probability $P(\xi_{t+1} = \eta_t)$ is the same as Algorithm 1. However, in the event $\{\xi_{t+1} \neq \eta_t\}$, $\xi_{t+1}$ and $\eta_t$ are independent in Algorithm 1 but correlated in Algorithm 3, achieved by the use of common random variates $\boldsymbol{Z}_2$ and $\boldsymbol{Z}_3$. The correlation between $\xi_{t+1}$ and $\eta_t$ helps to make $P(\xi_{t+2} = \eta_{t+1})$ larger, thus accelerating the meeting of $\{\xi_t\}$ and $\{\eta_{t-1}\}$. A more rigorous justification of this algorithm is given in Appendix A.2.

Finally, it is known that in the gradient expression (2), $f(\mathbf{x}; \boldsymbol{\theta}) = (\sigma(\boldsymbol{W}\boldsymbol{h} + \boldsymbol{b})\boldsymbol{h}^{\mathrm{T}}, \sigma(\boldsymbol{W}\boldsymbol{h} + \boldsymbol{b}), \boldsymbol{h})$ for RBM, corresponding to the parameters $\boldsymbol{\theta} = (\boldsymbol{W}, \boldsymbol{b}, \boldsymbol{c})$. With the coupled chains $\{\xi_t\}$ and $\{\eta_{t-1}\}$, RBM can then be trained using UCD given by Algorithm 2.

## 5 RELATED WORK

In this section we highlight the novelty of our article and clarify its overlap with prior art. In literature there were several attempts to prove the convergence of CD in special cases, or to reduce the bias of CD using other sampling techniques, all with undesirable results. For example, Yuille (2005) gave conditions for CD to converge, which unfortunately can hardly be satisfied in any realistic models. Jiang et al. (2018) showed a convergence result of CD for the exponential families, but consequently the model is restrictive and does not include the latent variable model. Krause et al. (2018) used importance sampling to estimate the normalizing constant, which is consistent with a large sample. However, it still induces a bias in the finite case, and the bias heavily depends on the choice of the importance weights. In contrast, the UCD method proposed in this article directly fixes the bias of CD, and hence bypasses the challenges in algorithm convergence.

Unbiased MCMC is a relatively new topic in statistics and machine learning. Some background knowledge in this article, for example Section 3.1, is taken from Jacob et al. (2017), which established a general framework for unbiased MCMC. Our new contributions are in the following aspects. First, we have developed Algorithm 1 and Theorem 1 exclusively for the Gibbs sampler, taking into account the special structure of Gibbs MCMC. Second, our theoretical results, including Theorem 1 and Corollary 2, have more practical assumptions than the ones in Jacob et al. (2017). For example, one of their key assumptions is that $\mathbb{E}\{|f(\xi_t)|^{2+c}\}$ is uniformly bounded for every finite step, which is quite abstract and hard to verify in practice compared with our Assumption 1. Third, Jacob et al. (2017) studied MCMC with a fixed target distribution, whereas we need to control the variance of estimators that evolve with parameter updates. Finally, in Section 4 we develop a specialized coupling algorithm for RBM, which is shown to be more efficient than the general one.

Another related work is the Markov chain Las Vegas method[1] (MCLV, Savarese et al., 2018), which also constructs an unbiased estimator for the score function of RBM. The main differences bewteen our method and MCLV are as follows: (1) MCLV is based on the regeneration theory of Markov chains, whereas UCD is built upon the coupling technique. (2) MCLV is exclusively designed for RBM, but UCD applies to a broader range of models. (3) In its current state, MCLV cannot handle continuous random variables, but UCD can. More comprehensive comparisons between MCLV and UCD are left for future exploration.

## 6 NUMERICAL EXPERIMENTS

### 6.1 BARS-AND-STRIPES DATA

We compare CD-$k$, PCD, and the proposed UCD algorithm for training RBM models on different data sets. In the first experiment we reproduce the results for the bars-and-stripes (BAS) data that have been studied by Schulz et al. (2010); Fischer & Igel (2010; 2014). It is a small data set with 36 data points and 16 binary variables, and is fit by a small model with 16 hidden units. However, it is one of the most important benchmark data sets for RBM since its log-likelihood value can be evaluated exactly, and it demonstrates the divergence of CD-based training algorithms. In our study, $k$ is set to 1 for CD (more experiments with larger $k$ are given in Appendix B.1), and each algorithm is run for 100 times, accounting for the randomness in the training process. A common learning

---

[1]We were informed of this method by a public comment on `https://openreview.net/forum?id=r1eyceSYPr`, after the current paper was accepted.

rate $\alpha = 0.01$ is set, and 1000 parallel Markov chains are used to approximate the gradient in each iteration. The results are shown in Figure 2.

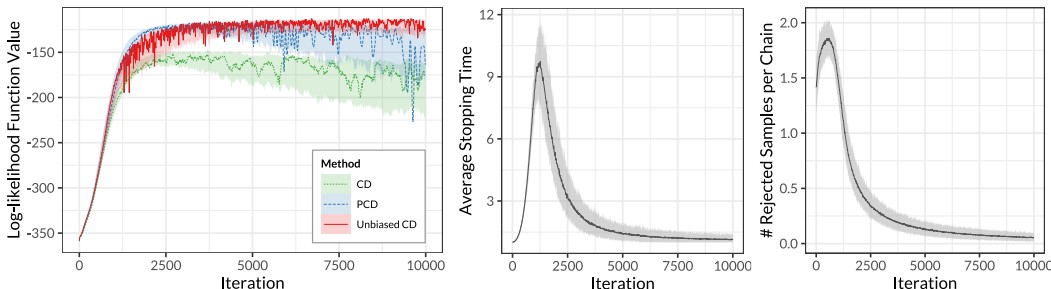

Figure 2: Left: exact log-likelihood values in each iteration. The shaded bands stand for the 2.5% and 97.5% quantiles across 100 runs, and the three trajectories in darker colors are sample learning curves in one run. Middle: average stopping time $\tau$ for UCD in each iteration. Right: average number of rejected samples in the coupling algorithm for UCD.

Figure 2 shows the following findings. First, we reproduce the results in Fischer & Igel (2014) that CD and PCD fail to converge to the true maximum likelihood value. In contrast, UCD does converge. Second, UCD has an adaptive choice of the stopping time in the Markov chain, compared to the fixed $k$ in CD. In the BAS data, the stopping time has a steep increase around the 1200th iteration. Interestingly, this is exactly where CD begins to fail. An interpretation of this phenomenon is that UCD automatically uses a large MCMC sample for parameter values that result in a "hard" distribution. An even more surprising fact is that the average stopping time $\tau$ for UCD is 2.40, making it computationally more efficient than the CD-20 algorithm, where 20 is the smallest $k$ such that CD-$k$ training is comparable to UCD (see Appendix B.1 for more discussions). Third, the cost of the rejection sampling step in UCD (line 7 of Algorithm 1) is tiny, as the number of rejected samples rarely goes above two. Finally, UCD does not see a massive increase in the variance. In fact, at the end of training the quantile band for UCD is much narrower than those of CD and PCD. All these findings further highlight the advantages of UCD.

## 6.2 SIMULATED RBM DATA

In the second example we show that the findings for the BAS data can be observed in other model settings. We simulate a data set from an RBM model with 200 visible units and 20 hidden units, where the entries of weight and bias parameters are all generated from a $\mathcal{N}(0, 1)$ distribution. The sample size of the simulated data set is 1000, and we fit an RBM model using 100 hidden units, which is larger than the true model since we intend to mimic the common practice of overparameterization in RBM training. We use a common learning rate $\alpha = 0.2$ and 1000 Markov chains in each iteration for all three algorithms. The log-likelihood values are approximated by Monte Carlo averages. The result is given in Figure 3, which shows similar patterns to the BAS data: CD and PCD eventually diverge, whereas UCD follows the typical behavior of SG. Additional experiments and discussions are given in Appendix B.2.

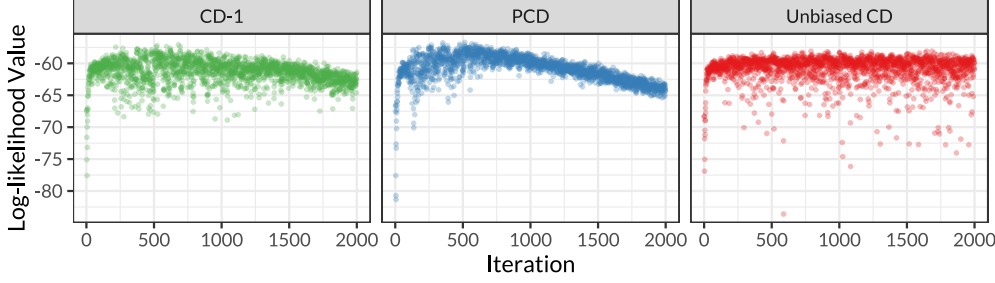

Figure 3: Approximate log-likelihood values for each algorithm on the simulated RBM data set.

### 6.3 FASHION-MNIST DATA

Next we consider the Fashion-MNIST data set[2], a replacement for the well-known but overused MNIST data set of handwritten digits (LeCun et al., 1990). Each data point in Fashion-MNIST contains 784 values within $[0, 1]$, representing a $28 \times 28$ greyscale image. The whole data set contains 60000 images, and we binarize the data by treating original values as probabilities and sampling from a Bernoulli distribution for each element. On the binarized data, we fit an RBM with 1000 hidden units, and train the model with different algorithms using a mini-batch size of 1000 and a learning rate $\alpha = 0.1$. For each training algorithm, 1000 parallel Markov chains are used to compute the gradient. Figure 4 demonstrates the training trajectories of the three algorithms.

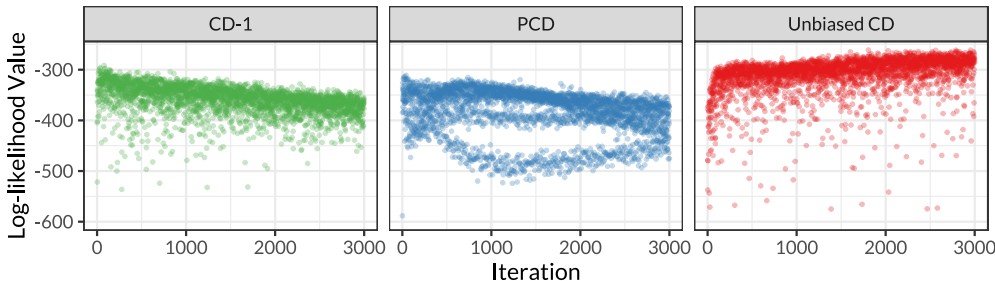

Figure 4: Approximate log-likelihood values for each algorithm on the Fashion-MNIST data set.

The training patterns for CD and PCD are surprising: the log-likelihood values of CD are decreasing, and PCD seems to bounce back and forth among three different paths. In fact, in Appendix B.3 we show that CD-$k$ does not converge even if $k$ is as large as 30, whereas the average stopping time for UCD is about 27. These results further demonstrate the superior performance of UCD.

## 7 DISCUSSION

In this article we use the unbiased MCMC technique to estimate the score function of energy-based latent variable models, which effectively fixes the bias of CD algorithms. Both the theoretical analysis and the numerical experiments at different scales demonstrate that the impact of bias elimination is huge.

It is expected that UCD may have slightly larger variance compared with CD and PCD, but we emphasize that the value of UCD is not a simple question of bias and variance trade-off. This is because for MCMC-based methods, the variance can always be reduced by running more independent Markov chains and taking the average, while removing the bias is highly non-trivial. Moreover, in the context of SG methods, bias is typically more harmful than variance, as the former may lead to divergence of the algorithm.

In terms of computational cost, it is true that UCD is in general slower than CD-1 or PCD, but it is comparable to or even faster than CD-$k$ that achieves a similar log-likelihood level. Therefore, UCD is not meant to completely replace CD or PCD, but rather to serve as an important addition to the existing training algorithms. In practice, a very useful technique is to first run the fast CD-1 or PCD to the near-optimum, and then proceed with UCD for guaranteed convergence.

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

# A    DISTRIBUTION COUPLING METHODS

## A.1    THE MAXIMAL COUPLING ALGORITHM

Let $(\xi, \eta)$ be a pair of random variables defined on the same probability space, and $p$ and $q$ be two distributions. $(\xi, \eta)$ is called a coupling of $p$ and $q$ if marginally $\xi \sim p$ and $\eta \sim q$. It is well known (see e.g. Proposition 4.7 of Levin & Peres, 2017) that among all possible joint distributions of $(\xi, \eta)$,

$$P(\xi = \eta) \le 1 - \|p - q\|_{\mathrm{TV}} = \int \min\{p(x), q(x)\}\mathrm{d}x, \qquad (4)$$

where $\|p - q\|_{\mathrm{TV}}$ is the total variation distance between $p$ and $q$. The maximal coupling algorithm, given in Algorithm 4, generates a coupling $(\xi, \eta)$ that achieves the bound in (4). That is, it maximizes the probability that two random variables are equal subject to their marginal distributions.

---

**Algorithm 4** Maximal coupling of two distributions $p(\cdot)$ and $q(\cdot)$, from Jacob et al. (2017).

---

**Input:** Density functions $p(\cdot)$ and $q(\cdot)$
**Output:** A coupling $(\xi, \eta)$ with $\xi \sim p(\cdot)$ and $\eta \sim q(\cdot)$
 1: Sample $\xi \sim p(\cdot)$ and $U \sim \mathsf{Uniform}(0, 1)$ independently
 2: **if** $U \le q(\xi)/p(\xi)$ **then**
 3:     **return** $(\xi, \xi)$
 4: **else**
 5:     Sample $\eta \sim q(\cdot)$ and $U' \sim \mathsf{Uniform}(0, 1)$ independently until $U' > p(\eta)/q(\eta)$
 6:     **return** $(\xi, \eta)$
 7: **end if**

---

## A.2    COUPLING METHOD FOR RBM

In the general maximal coupling method (Algorithm 4), let $p(\cdot) = \mathcal{T}(\cdot|\xi_t)$ and $q(\cdot) = \mathcal{T}(\cdot|\eta_{t-1})$ be the transition densities of RBM, and then it generates the new states $\xi_{t+1}$ and $\eta_t$ with the probability $P(\xi_{t+1} = \eta_t|\xi_t, \eta_{t-1})$ maximized. If the event $\{\xi_{t+1} = \eta_t\}$ does not happen, then $\xi_{t+1}$ and $\eta_t$ are sampled independently.

However, if $\xi_{t+1}$ and $\eta_t$ are close to each other, then in the next iteration the probability $P(\xi_{t+2} = \eta_{t+1}|\xi_{t+1}, \eta_t)$ would be large, which helps to shorten the stopping time. Therefore, we are motivated to minimize some type of distance between $\xi_{t+1}$ and $\eta_t$. For RBM, we characterize it by $\mathbb{E}(\|\mathbf{v}_{t+1} - \mathbf{v}'_t\|_2 | \mathbf{v}_{t+1} \ne \mathbf{v}'_t)$ and $\mathbb{E}(\|\mathbf{h}_{t+1} - \mathbf{h}'_t\|_2 | \mathbf{v}_{t+1}, \mathbf{v}'_t)$. Since all these variables are binary vectors, the norm of difference is basically the number of unequal components, and the problem reduces to the maximal coupling of Bernoulli variables.

First consider $\mathbb{E}(\|\mathbf{h}_{t+1} - \mathbf{h}'_t\|_2 | \mathbf{v}_{t+1}, \mathbf{v}'_t)$. Given the $\mathbf{v}$ variables, $\mathbf{h}_{t+1}$ and $\mathbf{h}'_t$ follow Bernoulli distributions elementwisely, with individual mean vectors denoted as $\mu_1$ and $\mu_2$, respectively. It can be shown that the maximal coupling of two Bernoulli variables is achieved by using the same random variate, so $\mathbb{E}(\|\mathbf{h}_{t+1} - \mathbf{h}'_t\|_2 | \mathbf{v}_{t+1}, \mathbf{v}'_t)$ is minimized by setting $\mathbf{h}_{t+1} = \mathbf{1}\{\mathbf{Z} \le \mu_1\}$ and $\mathbf{h}'_t = \mathbf{1}\{\mathbf{Z} \le \mu_2\}$, where $\mathbf{Z} \sim \mathsf{Uniform}([0, 1]^n)$. This leads to lines 15-16 of Algorithm 3.

Next, conditional on $\{\mathbf{v}_{t+1} \ne \mathbf{v}'_t\}$, the maximal coupling algorithm generates $\mathbf{v}_{t+1}$ and $\mathbf{v}'_t$ with marginal distributions proportional to $p_1(\boldsymbol{v}) - \min\{p_1(\boldsymbol{v}), p_2(\boldsymbol{v})\}$ and $p_2(\boldsymbol{v}) - \min\{p_1(\boldsymbol{v}), p_2(\boldsymbol{v})\}$, respectively, where $p_1(\cdot) = \mathcal{T}_v(\cdot|\boldsymbol{h}_t)$ and $p_2(\cdot) = \mathcal{T}_v(\cdot|\boldsymbol{h}'_{t-1})$. Algorithm 3 implements this by two rejection sampling steps (line 8 and line 11). If we use the common random variates $\boldsymbol{Z}_2 \sim \mathsf{Uniform}([0, 1]^n)$ for the proposals, then $\mathbf{v}_{t+1}$ and $\mathbf{v}'_t$ are maximally coupled in the event that they are accepted in the same iteration. This property helps to reduce $\mathbb{E}(\|\mathbf{v}_{t+1} - \mathbf{v}'_t\|_2 | \mathbf{v}_{t+1} \ne \mathbf{v}'_t)$.

To demonstrate that the specialized Algorithm 3 improves the general Algorithm 1, we sample coupled Markov chains from an RBM model with $m = 500$, $n = 100$, and the elements of $(\boldsymbol{W}, \boldsymbol{b}, \boldsymbol{c})$ are all generated from a $\mathcal{N}(0, 0.1^2)$ distribution. For both algorithms, the elements of the initial state $\boldsymbol{v}_0$ are sampled from independent Bernoulli(0.5) distributions, and we generate coupled Markov chains until the stopping time $\tau$ is reached or the chain length exceeds 1000. Figure 5 shows the distributions of $\tau$ for both algorithms based on 1000 replications.

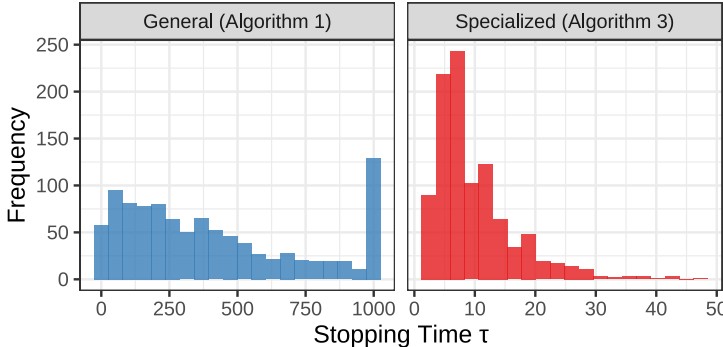

Figure 5: Distribution of the stopping time $\tau$ for the general (Algorithm 1) and specialized (Algorithm 3) coupling methods.

It is very clear that the general algorithm leads to a surprisingly long stopping time, and even has a long tail beyond 1000. In contrast, 65.4% of the stopping times in the specialized algorithm are smaller than or equal to 10. In this sense the improvement brought about by Algorithm 3 is huge.

## B    ADDITIONAL RESULTS FOR NUMERICAL EXPERIMENTS

All experiments in this article were run on an Intel® Xeon® Gold 6126 processor with 12 cores and 24 threads. CD and PCD algorithms used the OpenBLAS library[3] for parallel matrix computations, and UCD used OpenMP[4] to generate Markov chains in parallel.

### B.1    BAS DATA

For the BAS data, we gradually increase the value of $k$ in the CD-$k$ algorithm, and plot their training trajectories for the log-likelihood values (Figure 6). It can be seen that the smallest $k$ to make the result comparable to UCD is about 20. However, the stopping time for UCD has an average of 2.40 across all the iterations, which means that it is more efficient than a fixed-$k$ CD algorithm with a similar performance. The running time for each training algorithm, which is given in Table 1, also supports this claim.

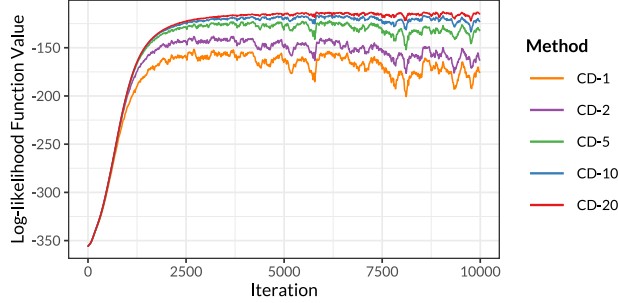

Figure 6: CD-$k$ algorithms for training the BAS data with different $k$.

### B.2    SIMULATED RBM DATA

For the experiment in Section 6.2, Figure 7 shows the training processes of CD-$k$ algorithms with larger $k$. Figure 8 gives the average stopping time for UCD in each iteration, and the number of

---
[3]https://www.openblas.net/
[4]https://www.openmp.org/

Table 1: Running time (in seconds) for each training algorithm on the BAS experiment.

| CD-1 w/ log-likelihood values | CD-1 | CD-2 | CD-5 | CD-10 | CD-20 | PCD | UCD |
|---|---|---|---|---|---|---|---|
| 65.01 | 5.38 | 8.44 | 17.89 | 33.17 | 63.51 | 5.18 | 15.99 |

discarded samples in the rejection sampling step. Table 2 illustrates the computational time for each algorithm. We can find that the computational cost of UCD is slightly larger than but very close to CD-1 and PCD.

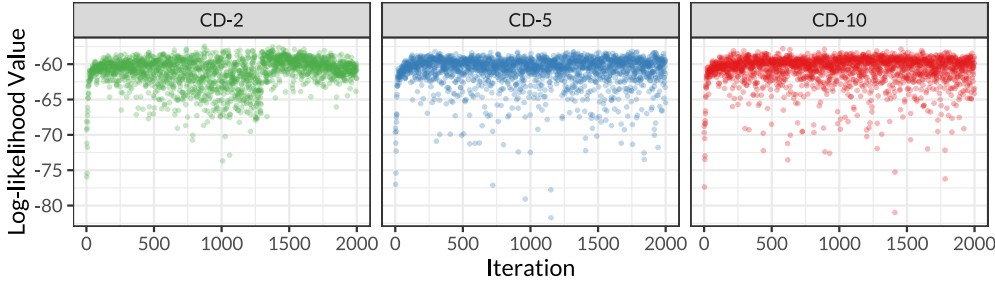

Figure 7: Approximate log-likelihood values for each algorithm on the simulated RBM data set.

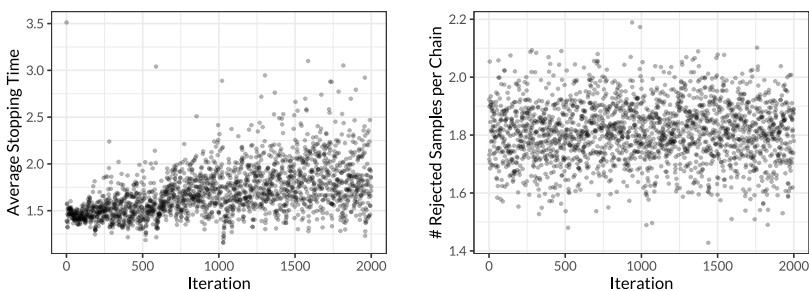

Figure 8: Left: average stopping time $\tau$ for UCD in training the model on simulated RBM data. Right: average number of rejected samples in the coupling algorithm for UCD.

Table 2: Running time (in seconds) for each training algorithm on the experiment in Section 6.2.

| CD-1 w/ log-likelihood values | CD-1 | CD-2 | CD-5 | CD-10 | PCD | UCD |
|---|---|---|---|---|---|---|
| 216.46 | 16.52 | 24.44 | 48.54 | 89.09 | 16.18 | 28.47 |

One useful technique to combine the computational efficiency of CD/PCD and the convergence of UCD is to use CD/PCD in the early stage of the optimization, and then take the resulting parameter values as initial starts for the UCD algorithm. Figure 9 demonstrates this idea: in the first 500 iterations the model is trained using PCD, and then the parameter values are fine-tuned by UCD with a small number of iterations and a reduced learning rate.

### B.3 FASHION-MNIST DATA

For the Fashion-MNIST data, Figure 10 gives the training trajectories of CD-10, CD-20, and CD-30 algorithms, but unfortunately none of them have a convergent pattern. Figure 11 shows the average stopping time and the number of discarded samples in the UCD algorithm. Again, the cost for rejection sampling can be ignored, and the value of $\tau$ gets stable around 30 after 2000 iterations.

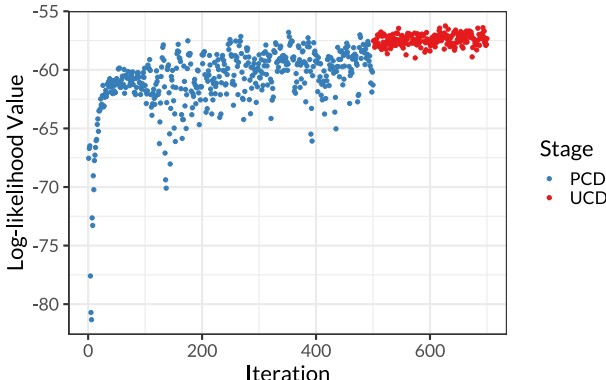

Figure 9: Combine PCD and UCD for efficient and convergent training.

Finally in Table 3 we demonstrate the running time for each training algorithm. Although UCD takes more time than other methods, the additional cost seems to be the necessary price to achieve a convergent result, as all other methods compared have a divergent log-likelihood.

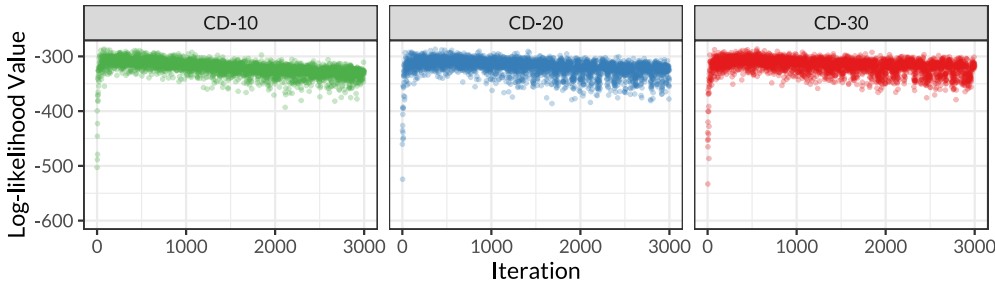

Figure 10: Approximate log-likelihood values for CD algorithms on the Fashion-MNIST data set.

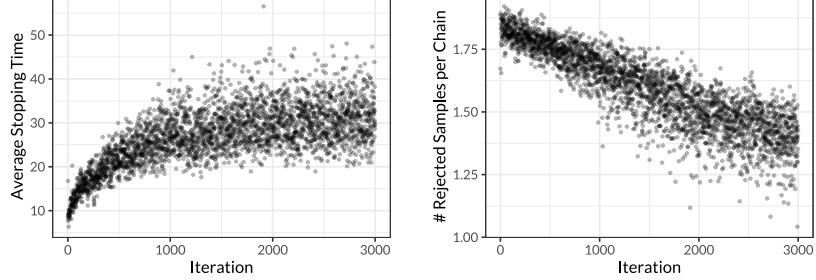

Figure 11: Left: average stopping time $\tau$ for UCD in training the model on Fashion-MNIST data. Right: average number of rejected samples in the coupling algorithm for UCD.

## C  VARIANCE OF GRADIENT ESTIMATES

To compare the variances of stochastic gradients generated by different algorithms, in each optimization iteration we estimate the following gradient variance defined by

$$V(\hat{f}(\boldsymbol{\theta})) = \mathbb{E}\left\{ \left\| \hat{f}(\boldsymbol{\theta}) - \mathbb{E}(\hat{f}(\boldsymbol{\theta})) \right\|_F^2 \right\},$$

Table 3: Running time (in minutes) for each training algorithm on the Fashion-MNIST experiment.

| CD-1 w/ log-likelihood values | CD-1 | CD-10 | CD-20 | CD-30 | PCD | UCD |
|---|---|---|---|---|---|---|
| 65.14 | 25.63 | 114.85 | 212.22 | 311.89 | 25.53 | 494.01 |

where $\hat{f}(\boldsymbol{\theta})$ is an estimator for the second term of (2), and $\|\cdot\|_F$ is the Frobenius norm. Among the training methods considered in this article, $\hat{f}(\boldsymbol{\theta})$ is biased in CD and PCD, and is unbiased in UCD. We consider the example in Section 6.2, and estimate $V(\hat{f}(\boldsymbol{\theta}))$ using the 1000 parallel Markov chains in each iteration. Figure 12 shows the estimated gradient variances for $\boldsymbol{b}$, $\boldsymbol{c}$, and $\boldsymbol{W}$ along the training process.

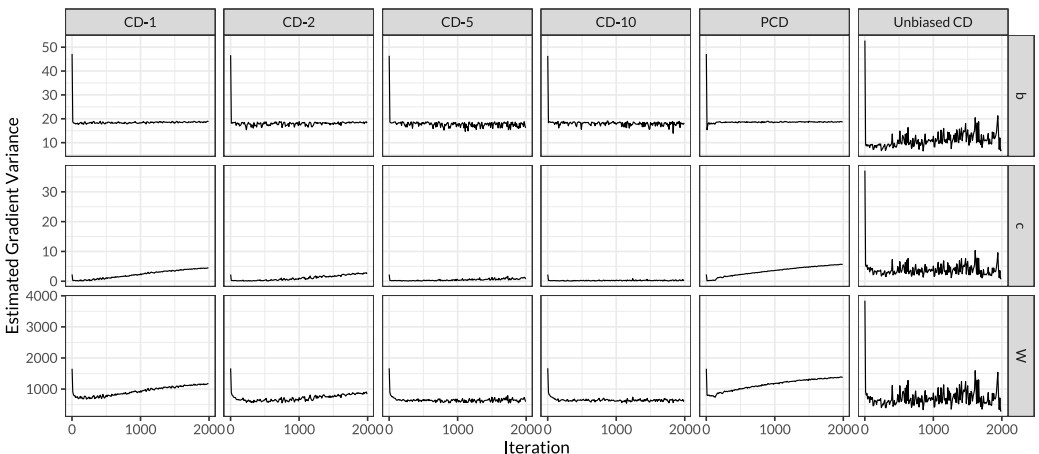

Figure 12: Estimated gradient variance in each optimization iteration for different methods.

It can be observed from Figure 12 that the gradient variance of UCD is generally comparable to CD and PCD, with only a few extreme cases. This finding suggests that the eliminated bias does not significantly increase the variance, which is important for the convergence speed of SG.

## D  PROOF OF THEOREMS

### D.1  THEOREM 1

Omitting the iteration index $i$ for brevity, we first prove that the stopping time $\tau$ has a finite expectation. Let $\mathcal{T}_{\boldsymbol{\theta}}(\boldsymbol{x}|\boldsymbol{x}') \coloneqq \mathcal{T}_{\boldsymbol{\theta}}(\boldsymbol{v}, \boldsymbol{h}|\boldsymbol{v}', \boldsymbol{h}') = p(\boldsymbol{v}|\boldsymbol{h}'; \boldsymbol{\theta})p(\boldsymbol{h}|\boldsymbol{v}; \boldsymbol{\theta})$ denote the transition density for a full update cycle. Under Assumptions 1 and 2, Lemma 2 of Johnson & Burbank (2015) shows that there exists a constant $\gamma \in [\gamma_1 \gamma_2, 1)$ such that

$$\mathbb{E}_{\mathbf{x} \sim \mathcal{T}_{\theta}(\boldsymbol{x}|\boldsymbol{x}')} l(\mathbf{h}) \leq \gamma l(\boldsymbol{h}') + \gamma_2 L_1 + L_2 \tag{5}$$

for all $\boldsymbol{x}' \in \mathbb{X}$. Therefore, the drift condition holds for the transition density $\mathcal{T}_{\boldsymbol{\theta}}(\boldsymbol{x}|\boldsymbol{x}')$.

Next, since Algorithm 1 is a maximal coupling algorithm, we have (see for example Jacob et al., 2017)

$$P(\xi_{t+1} = \eta_t|\xi_t = \boldsymbol{x}, \eta_{t-1} = \boldsymbol{x}') = \int \min\{\mathcal{T}_{\boldsymbol{\theta}}(\boldsymbol{y}|\boldsymbol{x}), \mathcal{T}_{\boldsymbol{\theta}}(\boldsymbol{y}|\boldsymbol{x}')\} \mathrm{d}\boldsymbol{y}.$$

By Assumption 2, $p(\boldsymbol{v}|\boldsymbol{h}; \boldsymbol{\theta}) \geq \varepsilon q(\boldsymbol{v})$ for all $\boldsymbol{h} \in \mathbb{D}$, $\boldsymbol{v} \in \mathbb{V}$, and $\boldsymbol{\theta} \in \Theta$, so

$$P(\xi_{t+1} = \eta_t|\xi_t = \boldsymbol{x}, \eta_{t-1} = \boldsymbol{x}') \geq \int \varepsilon q(\tilde{\boldsymbol{v}})p(\tilde{\boldsymbol{h}}|\tilde{\boldsymbol{v}}; \boldsymbol{\theta}) \mathrm{d}\boldsymbol{y}$$

on $\boldsymbol{x}, \boldsymbol{x}' \in \mathbb{D} \times \mathbb{V}$, where $\boldsymbol{y} = (\tilde{\boldsymbol{v}}, \tilde{\boldsymbol{h}})$. Note that $q(\tilde{\boldsymbol{v}})p(\tilde{\boldsymbol{h}}|\tilde{\boldsymbol{v}}; \boldsymbol{\theta})$ is a joint density function, so $\int q(\tilde{\boldsymbol{v}})p(\tilde{\boldsymbol{h}}|\tilde{\boldsymbol{v}}; \boldsymbol{\theta}) \mathrm{d}\boldsymbol{y} = 1$, and hence $P(\xi_{t+1} = \eta_t|\xi_t = \boldsymbol{x}, \eta_{t-1} = \boldsymbol{x}') \geq \varepsilon$ for all $\boldsymbol{h} \in \mathbb{D}$ and $\boldsymbol{\theta} \in \Theta$.

Then by Proposition 3.4 of Jacob et al. (2017), there exist constants $\kappa_1 > 0$ and $\rho_1 \in (0, 1)$ such that for all $t > 0$, $P(\tau > t) \le \kappa_1 l(\boldsymbol{h}_0) \rho_1^t$. As a result, we obtain

$$\mathbb{E}(\tau) = \sum_{t=0}^{\infty} P(\tau > t) \le \frac{\kappa_1 l(\boldsymbol{h}_0)}{1 - \rho_1}. \tag{6}$$

The right hand side of (6) does not depend on the value of $\boldsymbol{\theta}$, so $\mathbb{E}(\tau)$ is uniform in the iteration index $i$.

To make sure that $\tilde{g}(\boldsymbol{\theta})$ is well defined, a few regularity conditions, formulated as Assumption 2.1 of Jacob et al. (2017), need to be verified. Under the drift condition (5) and Assumption 2, Theorem 12 of Rosenthal (1995) shows that $\{\xi_t\}$ is a geometrically ergodic Markov chain. Therefore, there exist constants $\kappa_2 > 0$ and $\rho_2 \in (0, 1)$ such that

$$|\mathbb{E}\phi(\xi_t) - \mathbb{E}_{\mathcal{M}}\phi| \le \kappa_2 l(\boldsymbol{h}_0)\rho_2^t \tag{7}$$

for all $\phi : |\phi(\boldsymbol{v}, \boldsymbol{h})| \le l(\boldsymbol{h})$. Therefore, there exists a constant $M > 0$ such that $|\mathbb{E}\phi(\xi_t)| \le \mathbb{E}_{\mathcal{M}} l + \kappa_2 l(\boldsymbol{h}_0)\rho_2^t \le D + \kappa_2 l(\boldsymbol{h}_0) \le M < \infty$.

Since $|f(\boldsymbol{x}; \boldsymbol{\theta})|^{2+c} \le l(\boldsymbol{h})$ by Assumption 1, we get $\mathbb{E}\{|f(\xi_t)|^{2+c}\} \le M$ for all $t > 0$. Moreover, (7) implies that $\mathbb{E}\{f(\xi_t)\} \to \mathbb{E}_{\mathcal{M}} f$. These two results verify Assumption 2.1 of Jacob et al. (2017), and hence the unbiasedness of $\tilde{g}(\boldsymbol{\theta})$ is true by design.

Finally, we need to show that the second moment of the stochastic gradient is bounded uniformly in $\boldsymbol{\theta}$. Let $\Delta_k = f(\xi_k)$ and $\Delta_t = f(\xi_t) - f(\eta_{t-1})$ for $t \ge k + 1$, and then $\tilde{g}_2(\boldsymbol{\theta}) = \sum_{t=k}^{\infty} \Delta_t$. From the ergodicity property (7) we immediately get $\mathbb{E}(\Delta_k^2) \le M$. For $t \ge k + 1$,

$$\mathbb{E}(|\Delta_t|^{2+c}) = \mathbb{E}(|f(\xi_t) - f(\eta_{t-1})|^{2+c}) \le 2^{1+c}\mathbb{E}(|f(\xi_t)|^{2+c} + |f(\eta_{t-1})|^{2+c}) \le 2^{2+c}M.$$

Then by Hölder's inequality,

$$\mathbb{E}(\Delta_t^2) = \mathbb{E}(\Delta_t^2 \mathbf{1}\{\tau > t\}) \le \left\{\mathbb{E}(|\Delta_t|^{2+c})\right\}^{2/(c+2)} \left\{P(\tau > t)\right\}^{c/(c+2)}$$

$$\le 4M^{2/(c+2)} \left\{\kappa_1 l(\boldsymbol{h}_0)\rho_1^t\right\}^{c/(c+2)}.$$

Since $\tilde{g}_2(\boldsymbol{\theta}) = \sum_{t=k}^{\infty} \Delta_t$, we get

$$\mathbb{E}\left[\{\tilde{g}_2(\boldsymbol{\theta}) - \Delta_k\}^2\right] = \mathbb{E}\left\{\left(\sum_{t=k+1}^{\infty} \Delta_t\right)^2\right\} \le \sum_{t=k+1}^{\infty}\sum_{s=k+1}^{\infty} \mathbb{E}|\Delta_t\Delta_s \mathbf{1}\{\tau > t\}\mathbf{1}\{\tau > s\})|.$$

We also have

$$\mathbb{E}|\Delta_t\Delta_s \mathbf{1}\{\tau > t\}\mathbf{1}\{\tau > s\})| \le \sqrt{\mathbb{E}(\Delta_t^2 \mathbf{1}\{\tau > t\})\mathbb{E}(\Delta_s^2 \mathbf{1}\{\tau > s\})} \le C_0 \rho_3^s \rho_3^t$$

for some constants $C_0 > 0$ and $\rho_3 \in (0, 1)$, so finally,

$$\mathbb{E}\left[\{\tilde{g}_2(\boldsymbol{\theta})\}^2\right] \le 2\mathbb{E}(\Delta_k^2) + 2\mathbb{E}\left[\{\tilde{g}_2(\boldsymbol{\theta}) - \Delta_k\}^2\right] \le 2M + 2\sum_{t=k+1}^{\infty}\sum_{s=k+1}^{\infty} C_0 \rho_3^s \rho_3^t < \infty,$$

and the bound does not depend on $\boldsymbol{\theta}$.

## D.2 COROLLARY 2

Since $E(\boldsymbol{v}, \boldsymbol{h}; \boldsymbol{\theta})$ is continuous in $\boldsymbol{\theta}$, every conditional distribution of $p(\boldsymbol{v}, \boldsymbol{h}; \boldsymbol{\theta})$ is also continuous in $\boldsymbol{\theta}$. This implies that $\mathcal{T}_{\boldsymbol{\theta}}(\boldsymbol{x}|\boldsymbol{x}')$ is continuous in $\boldsymbol{\theta}$ as well. We have assumed that $\mathcal{T}_{\boldsymbol{\theta}}$ is irreducible and aperiodic, so for each $\boldsymbol{\theta}$, $\mathcal{T}_{\boldsymbol{\theta}}(\boldsymbol{x}|\boldsymbol{x}') > 0$ for every $\boldsymbol{x}, \boldsymbol{x}' \in \mathbb{X}$. By the compactness of $\Theta$, there exist a constant $\varepsilon > 0$ such that $\mathcal{T}_{\boldsymbol{\theta}}(\boldsymbol{x}|\boldsymbol{x}') \ge \varepsilon$ for all $\boldsymbol{x}, \boldsymbol{x}' \in \mathbb{X}$ and $\boldsymbol{\theta} \in \Theta$.

Similarly, $f(\boldsymbol{x}; \boldsymbol{\theta}) = \partial E(\boldsymbol{v}, \boldsymbol{h}; \boldsymbol{\theta})/\partial\boldsymbol{\theta}$ is continuous in $\boldsymbol{\theta}$, so for any $c > 0$, there is a constant $M > 1$ such that $|f(\boldsymbol{x}; \boldsymbol{\theta})|^{2+c} \le M$ for all $\boldsymbol{x} \in \mathbb{X}$ and $\boldsymbol{\theta} \in \Theta$. Then by choosing constant functions $l(\boldsymbol{h}) = M$, $r(\boldsymbol{v}) = M$ and constants $\gamma_1 = \gamma_2 = 1/2$ and $L_1 = L_2 = M/2$, we make Assumption 1 hold.

For Assumption 2, the density function $q(\cdot)$ can be chosen as a uniform distribution over the finite space $\mathbb{X}$. Then the proof is complete.

