# OpenReview forum: "Unbiased Contrastive Divergence Algorithm for Training Energy-Based Latent Variable Models"
_ICLR.cc/2020/Conference — Accept (Spotlight)_

### Official Review · AnonReviewer1 · 2019-10-21
**Official Blind Review #1**

**Rating:** 8

**Review:**

The paper proposes an algorithmic improvement that significantly simplifies training of energy-based models, such as the Restricted Boltzmann Machine. The key issue in training such models is computing the gradient of the log partition function, which can be framed as computing the expected value of f(x) = dE(x; theta) / d theta over the model distribution p(x). The canonical algorithm for this problem is Contrastive Divergence which approximates x ~ p(x) with k steps of Gibbs sampling, resulting in biased gradients. In this paper, the authors apply the recently introduced unbiased MCMC framework of Jacob et al. to completely remove the bias. The key idea is to (1) rewrite the expectation as a limit of a telescopic sum: E f(x_0) + \sum_t E f(x_t) - E f(x_{t-1}); (2) run two coupled MCMC chains, one for the “positive” part of the telescopic sum and one for the “negative” part until they converge. After convergence, all remaining terms of the sum are zero and we can stop iterating. However, the number of time steps until convergence is now random.

Other contributions of the paper are:
1. Proof that Bernoulli RBMs and other models satisfying certain conditions have finite expected number of steps and finite variance of the unbiased gradient estimator.
2. A shared random variables method for the coupled Gibbs chains that should result in faster convergence of the chains.
3. Verification of the proposed method on two synthetic datasets and a subset of MNIST, demonstrating more stable training compared to contrastive divergence and persistent contrastive divergence.

I am very excited about this paper and strongly support its acceptance, since the proposed method should revitalize research in energy-based models. While I find the experiments to be somewhat lacking, this is sufficiently offset by the theoretical contributions of the paper.

Pros
1. The paper reads well and introduces all the necessary preliminaries to understand the method. This is important, since I expect many readers to be unfamiliar with the technique.
2. The proposed method solves an important problem which, as far as I understand, has been the roadblock in large-scale training of RBMs and related models. It is also elegant and fairly straightforward to implement.
3. The proof of finite computation time and variance is very nice to have. This is because in some cases removing the bias leads to infinite variance, e.g. a parallel submission on SUMO (https://openreview.net/forum?id=SylkYeHtwr).

Cons
1. I don’t think Corollary 1 (convergence of gradient descent to the global optimum) is true for RBMs, as stated on Page 6. This is because the log-likelihood of RBM, or indeed any latent-variable model with permutation-invariant latents, is non-convex. I would suggest removing this corollary and simplifying Algorithm 2 to be regular SGD, as used in the experiments.
2. There is no experimental comparison of Algorithm 1 (the general version) and Algorithm 3 (the specialized RBM version). It seems intuitive that the specialized version should have lower computation time, but this must be confirmed.
3. The experimental section may be significantly improved.
* It is unclear what value of k (number of initial Gibbs steps) from Algorithm 2 is used.
* The experiments on just the “0” digits of MNIST seem a bit simplistic for the year 2019. It is also not clear what binarization protocol is used.
* It would be very helpful to provide estimates of the gradient (not log-likelihood) variance of each method to better understand the trade-off between the bias and the variance.
* I would also like to see the wall-clock time comparison of the methods.

Minor comments
* Page 1. Of this kind -> of this class. The data distribution p_v (v; theta) -> The model distribution
* Page 2. Property -> properties. CD-\tau -- I don’t think you can correctly refer to your method in this way, since it has at least double the computation time of CD for the same number of iterations.
* Page 3. Provides -> provide. Likelihood gradient -> log-likelihood gradient
* Algorithm 1 is an infinite loop with no break clause. It would be good to add a break statement after line 5. This would also simplify the discussion of the method.
* Page 7. I wouldn’t call the fact that CD doesn’t converge on the BAS dataset remarkable, given that it’s been reported by Fischer & Igel 2014.
* Page 9. The last paragraph stating that the proposed method is not a replacement for CD is confusing. Can you add a short experiment to demonstrate that this combination makes sense?

**Experience Assessment:**

I have read many papers in this area.

**Review Assessment: Checking Correctness Of Derivations And Theory:**

I assessed the sensibility of the derivations and theory.

**Review Assessment: Checking Correctness Of Experiments:**

I carefully checked the experiments.

**Review Assessment: Thoroughness In Paper Reading:**

I read the paper thoroughly.

---

> ### Author Response · Authors · 2019-11-15
> **Improved experiments and various fixes**
>
> Thanks for the helpful comments and corrections. We have included many new numerical results in the updated manuscript, and below are our line-to-line responses.
>
> >>> 1. I don’t think Corollary 1 (convergence of gradient descent to the global optimum) is true for RBMs, as stated on Page 6. This is because the log-likelihood of RBM, or indeed any latent-variable model with permutation-invariant latents, is non-convex. I would suggest removing this corollary and simplifying Algorithm 2 to be regular SGD, as used in the experiments.
>
> For Theorem 1 and Algorithm 2 we do not assume a specific model for $p(v,h)$, and Corollary 1 is mainly used to demonstrate a typical convergence result for SGD. Indeed it does not apply to RBM as the objective function is not convex, so on top of page 6 we have mentioned that there are other versions of the theorem, for different types of objective functions. We have made this clearer in the updated manuscript.
>
> >>> 2. There is no experimental comparison of Algorithm 1 (the general version) and Algorithm 3 (the specialized RBM version). It seems intuitive that the specialized version should have lower computation time, but this must be confirmed.
>
> Thanks for pointing out. We have added this comparison in Appendix A.2.
>
> >>> 3. The experimental section may be significantly improved.
> >>> * It is unclear what value of k (number of initial Gibbs steps) from Algorithm 2 is used.
>
> In all our experiments we set k=1. We have added this point to the text.
>
> >>> * The experiments on just the “0” digits of MNIST seem a bit simplistic for the year 2019. It is also not clear what binarization protocol is used.
>
> We have significantly increased the size of the experiment: a full Fashion-MNIST data set with n=1000 hidden units. We treat data values as probabilities and globally binarize all the data points by sampling from Bernoulli distributions. The binarized data are then passed to the models.
>
> >>> * It would be very helpful to provide estimates of the gradient (not log-likelihood) variance of each method to better understand the trade-off between the bias and the variance.
>
> We have included such an analysis in Appendix C.
>
> >>> * I would also like to see the wall-clock time comparison of the methods.
>
> We have included the timing comparisons in Appendix B.
>
> >>> Minor comments
> >>> * Page 1. Of this kind -> of this class. The data distribution p_v (v; theta) -> The model distribution
>
> Corrected.
>
> >>> * Page 2. Property -> properties. CD-\tau -- I don’t think you can correctly refer to your method in this way, since it has at least double the computation time of CD for the same number of iterations.
>
> We have removed this notation and changed the wording.
>
> >>> * Page 3. Provides -> provide. Likelihood gradient -> log-likelihood gradient
>
> Corrected.
>
> >>> * Algorithm 1 is an infinite loop with no break clause. It would be good to add a break statement after line 5. This would also simplify the discussion of the method.
>
> We have added a maximum stopping time to Algorithm 1.
>
> >>> * Page 7. I wouldn’t call the fact that CD doesn’t converge on the BAS dataset remarkable, given that it’s been reported by Fischer & Igel 2014.
>
> Fixed and rewritten.
>
> >>> * Page 9. The last paragraph stating that the proposed method is not a replacement for CD is confusing. Can you add a short experiment to demonstrate that this combination makes sense?
>
> Yes, we have added an example in Appendix B.2 and Figure 9.

---

### Official Review · AnonReviewer2 · 2019-10-21
**Official Blind Review #2**

**Rating:** 8

**Review:**

The paper introduces an efficient, unbiased contrastive divergence-like algorithm for training energy-based generative models on the example of Restricted Boltzmann Machine.
The proposed algorithm is built upon a very interesting work on unbiased finite-step MCMC approximations by Jacob et al, 2017.
Despite the actual theory being published some time ago, the submitted paper popularises these ideas in the machine learning community and contains optimised variants of the existing algorithms for training of RBMs.

The paper is mostly written well and does a good job of introducing unbiased MCMC estimators.
Authors evaluate their method on rather toyish datasets (by modern standards), however, their empirical analysis is thorough. The improvement upon the standard CD and persistent CD is clear.
It also appears that the algorithm actually does not require too many steps and generally does not introduce a lot of computational overhead.
The only question I have is why CD has only been tried with k=1 steps?
I would be interested in its performance for different number of steps including the dynamically chosen number provided by the empirical \tau in UCD for a given iteration.
Even though I do not expect a significant improvement to be obtained, this would separate the effect of the number of steps chosen “right” from unbiasedness of the gradient estimator.
Other baselines, including those mentioned in the related work, could also make the comparison more complete.

I would also suggest including https://arxiv.org/abs/1905.04062, as it seems to be relevant in the spirit.

**Experience Assessment:**

I have read many papers in this area.

**Review Assessment: Checking Correctness Of Derivations And Theory:**

I assessed the sensibility of the derivations and theory.

**Review Assessment: Checking Correctness Of Experiments:**

I carefully checked the experiments.

**Review Assessment: Thoroughness In Paper Reading:**

I read the paper at least twice and used my best judgement in assessing the paper.

---

> ### Author Response · Authors · 2019-11-15
> **Improved numerical experiments**
>
> Thanks for the suggestions. We have added more comparisons in the new version.
>
> >>> The only question I have is why CD has only been tried with k=1 steps?
> I would be interested in its performance for different number of steps including the dynamically chosen number provided by the empirical \tau in UCD for a given iteration.
> Even though I do not expect a significant improvement to be obtained, this would separate the effect of the number of steps chosen “right” from unbiasedness of the gradient estimator.
> Other baselines, including those mentioned in the related work, could also make the comparison more complete.
>
> We have significantly improved the numerical experiments with larger models, and have included CD-k algorithms with larger k in Appendix B.
>
> >>> I would also suggest including https://arxiv.org/abs/1905.04062, as it seems to be relevant in the spirit.
>
> We have added this article to our reference.

---

### Official Review · AnonReviewer3 · 2019-10-23
**Official Blind Review #3**

**Rating:** 6

**Review:**

Based on recent progress in unbiased MCMC sampling the paper proposes an unbiased contrastive divergence (UCD) algorithm for training energy based models. Specifically they developed an unbiased version of the gibbs sampling contrastive divergence algorithm for training restricted Boltzman machines. The authors demonstrate their method on a toy dataset, simulated data, as well as a reduced version (only the zero digits) of the MNIST dataset and compare the results with the standard Contrastive divergence and Persistent Contrastive Divergence methods.

Score:
I find the line of work on unbiased estimators important and the (although i’m not an expert) the theory in the paper seems sound. Further the paper is well written and relatively easy to follow. However I do not find the experimental section completely comprehensive and some of the results seem to achieve worse performance than what is reported in the litterature for both the proposed method and baselines (see detailed questions below). Overall I currently score the paper as a weak reject although I can be convinced to bump the score depending on the author feedback.

Detailed Questions:
Experimental Results:
Q1) In [Tieleman2008] log-likelihood values for the full MNIST dataset using a) a small model (25 hidden units) where the likelihood is computed exactly and b) an bigger model (500 hidden units) where the likelihood is approximated. On the full MNIST dataset they train using PCD, CD-1, CD-10 and report approximately Log-Likelihoods of -130 and -85 for the small and large models respectively. My questions are:
Q1.1) In figure 4 you report approximate log-likelihood values on MNIST (only digits zero) of -150 for the different samplers using an RBM with100 hidden units. That seems to be lower performance than the models in [Tieleman2008] while training on a presumably easier dataset?

Q1.2) In figure 4. Can you comment a bit on the variance of your method which seems to be higher, Is there a Bias/Variance trade-off between UCD and e.g PCD?

Q1.3) [Tieleman2008] Reports training times of 1 to 9 Hours for training in on the full MNIST dataset in 2008 and [Hinton 2006] trained large RBMs in 2006. Why is that setting then computationally time-consuming today in your setup - Is there some difference in the setup that I'm missing?

Q1.4) I highly value enlightening small scale experiments and do understand that computational resources are not available everywhere however I think it would benefit the paper greatly if the proposed method is demonstrated on some reasonably sized dataset (at the very least one of full MNIST, Fashion MNIST, FreyFaces).

Q1.5) In Figure 2 you show some interesting figures for the average stopping time and number of rejected samples on the BAS toy dataset. How does these results look on a real dataset like the MNIST zero digit data?

[Tieleman 2008], Training Restricted Boltzmann Machines using Approximations to the Likelihood Gradient,
[Hinton 2006] Reducing the Dimensionality of Data with Neural Networks


**Experience Assessment:**

I have read many papers in this area.

**Review Assessment: Checking Correctness Of Derivations And Theory:**

I did not assess the derivations or theory.

**Review Assessment: Checking Correctness Of Experiments:**

I carefully checked the experiments.

**Review Assessment: Thoroughness In Paper Reading:**

I read the paper at least twice and used my best judgement in assessing the paper.

---

> ### Author Response · Authors · 2019-11-15
> **Increased size of data set and model**
>
> Thanks for the constructive comments on numerical experiments, and we have adopted the suggestion to compute on a large data set (the full Fashion-MNIST) with a large model (1000 hidden units). We have also included many other discussions such as the variance of UCD and the computational cost. The detailed responses are as follows.
>
> >>> Q1) In [Tieleman2008] log-likelihood values for the full MNIST dataset using a) a small model (25 hidden units) where the likelihood is computed exactly and b) an bigger model (500 hidden units) where the likelihood is approximated. On the full MNIST dataset they train using PCD, CD-1, CD-10 and report approximately Log-Likelihoods of -130 and -85 for the small and large models respectively. My questions are:
> >>> Q1.1) In figure 4 you report approximate log-likelihood values on MNIST (only digits zero) of -150 for the different samplers using an RBM with100 hidden units. That seems to be lower performance than the models in [Tieleman2008] while training on a presumably easier dataset?
>
> In general the likelihood values are not comparable with different data sets. In the updated version we have trained a much larger model (1000 hidden units) with the full Fashion-MNIST data. We hope the new experiment is more convincing.
>
> >>> Q1.2) In figure 4. Can you comment a bit on the variance of your method which seems to be higher, Is there a Bias/Variance trade-off between UCD and e.g PCD?
>
> Yes, in the updated version we discuss the variance of UCD in Appendix C.
>
> >>> Q1.3) [Tieleman2008] Reports training times of 1 to 9 Hours for training in on the full MNIST dataset in 2008 and [Hinton 2006] trained large RBMs in 2006. Why is that setting then computationally time-consuming today in your setup - Is there some difference in the setup that I'm missing?
>
> What we meant about "time-consuming" is the following: we found that the MNIST data set was quite "benign", or "robust", in the sense that even biased algorithms such as CD can train a reasonably good model. Therefore, it may take a very long time to actually observe the divergence of CD (recall that even in the small BAS data, it takes thousands of iterations). But on the Fashion-MNIST data that we use in the updated manuscript, it is easy to see the differences of CD, PCD, and UCD, even with a small number of iterations.
>
> >>> Q1.4) I highly value enlightening small scale experiments and do understand that computational resources are not available everywhere however I think it would benefit the paper greatly if the proposed method is demonstrated on some reasonably sized dataset (at the very least one of full MNIST, Fashion MNIST, FreyFaces).
>
> Thanks for the recommendation. We have updated our experiment based on the full Fashion-MNIST data.
>
> >>> Q1.5) In Figure 2 you show some interesting figures for the average stopping time and number of rejected samples on the BAS toy dataset. How does these results look on a real dataset like the MNIST zero digit data?
>
> In fact we have included such results in Appendix B. In the new manuscript the plot for Fashion-MNIST is in Figure 11.

---

### Public Comment · ~Jianwen_Xie1 · 2020-01-08
**related work about EBM using neural nets**

Excellent work !

Might have been nice to cite and discuss some prior work about EBMs using neural nets as energy functions, such as

[1] A Theory of Generative ConvNet.
Jianwen Xie *, Yang Lu *, Song-Chun Zhu, Ying Nian Wu (ICML 2016)

[2] Synthesizing Dynamic Pattern by Spatial-Temporal Generative ConvNet
Jianwen Xie, Song-Chun Zhu, Ying Nian Wu (CVPR 2017)

[3] Learning Descriptor Networks for 3D Shape Synthesis and Analysis
Jianwen Xie *, Zilong Zheng *, Ruiqi Gao, Wenguan Wang, Song-Chun Zhu, Ying Nian Wu (CVPR) 2018

[4]  Learning generative ConvNets via multigrid modeling and sampling.
R Gao*, Y Lu*, J Zhou, SC Zhu, and YN Wu (CVPR 2018).

[5] On learning non-convergent non-persistent short-run MCMC toward energy-based model.
E Nijkamp, M Hill, SC Zhu, and YN Wu (NeurIPS 2019)

Thanks.

---

> ### Author Response · Authors · 2020-01-31
> **References added**
>
> Thank you Jianwen. It took me a while to read and absorb these papers, and now they have been added.

---

> > ### Public Comment · ~Jianwen_Xie1 · 2020-04-30
> > **Thanks**
> >
> > Thank you so much !

---

### Public Comment · ~Mayank_Kakodkar1 · 2020-02-08
**Great Work!**

I really enjoyed reading your paper!

Could you please consider citing [1].
[1] proposed a method called Markov Chain Las Vegas(MCLV), which computes unbiased estimates of the RBM gradient.
MCLV uses random walk tours that begin and end at a 'supernode', which is an aggregation of high probability states called constructed using the training data. [1] also proposed a biased version of this estimator called MCLV-K which considers tours of lengths upto K and showed both theoretically and empirically that since the tour length distribution has a geometrically decaying tail, the bias is minimal.

[1] Pedro Savarese, (Mayank Kakodar), Bruno Ribeiro, From Monte Carlo to Las Vegas: Improving Restricted Boltzmann Machine Training through Stopping Sets, AAAI, 2018 [arXiv:1711.08442]

---

> ### Author Response · Authors · 2020-02-16
> **A relevant and interesting paper**
>
> Thank you, Mayank. This is a very relevant and interesting paper, and I wish I could see it earlier. I will cite and add some discussions in a revision later.

---

> > ### Public Comment · ~Mayank_Kakodkar1 · 2020-02-19
> > **Thanks!**
> >
> > Thank you so much!

---

### Decision · Program_Chairs · 2019-12-19

**Decision:**

Accept (Spotlight)

**Comment:**

Main content:

Blind review #1 summarizes it well:

The paper proposes an algorithmic improvement that significantly simplifies training of energy-based models, such as the Restricted Boltzmann Machine. The key issue in training such models is computing the gradient of the log partition function, which can be framed as computing the expected value of f(x) = dE(x; theta) / d theta over the model distribution p(x). The canonical algorithm for this problem is Contrastive Divergence which approximates x ~ p(x) with k steps of Gibbs sampling, resulting in biased gradients. In this paper, the authors apply the recently introduced unbiased MCMC framework of Jacob et al. to completely remove the bias. The key idea is to (1) rewrite the expectation as a limit of a telescopic sum: E f(x_0) + \sum_t E f(x_t) - E f(x_{t-1}); (2) run two coupled MCMC chains, one for the “positive” part of the telescopic sum and one for the “negative” part until they converge. After convergence, all remaining terms of the sum are zero and we can stop iterating. However, the number of time steps until convergence is now random.

Other contributions of the paper are:
1. Proof that Bernoulli RBMs and other models satisfying certain conditions have finite expected number of steps and finite variance of the unbiased gradient estimator.
2. A shared random variables method for the coupled Gibbs chains that should result in faster convergence of the chains.
3. Verification of the proposed method on two synthetic datasets and a subset of MNIST, demonstrating more stable training compared to contrastive divergence and persistent contrastive divergence.

--

Discussion:

The main objection in reviews was to have meaningful empirical validation of the strong theoretical aspect of the paper, which the authors did during the rebuttal period to the satisfaction of reviewers.

--

Recommendation and justification:

As review #1 said, "I am very excited about this paper and strongly support its acceptance, since the proposed method should revitalize research in energy-based models."